# Signaling Lymphocytic Activation Molecule Family Member 1 Inhibits Porcine Reproductive and Respiratory Syndrome Virus Replication

**DOI:** 10.3390/ani12243542

**Published:** 2022-12-15

**Authors:** Haofei Song, Jima Quan, Chang Li, Wan Liang, Lan Zhang, Shuangshuang Wang, Hongyu Lu, Keli Yang, Danna Zhou, Peng Li, Yongxiang Tian

**Affiliations:** 1College of Animal Science, Yangtze University, Jingzhou 434025, China; 2Key Laboratory of Prevention and Control Agents for Animal Bacteriosis (Ministry of Agriculture and Rural Affairs), Hubei Provincial Key Laboratory of Animal Pathogenic Microbiology, Institute of Animal Husbandry and Veterinary, Hubei Academy of Agricultural Sciences, Wuhan 430064, China

**Keywords:** PRRSV, SLAMF1, replication, innate immunity, inflammatory cytokines

## Abstract

**Simple Summary:**

PRRS is one of the most important diseases that has brought significant economic losses to the swine industry worldwide. SLAMF1 is a costimulatory factor that is involved in innate immunity, inflammation, and infection. In this study, we demonstrate that overexpression of the SLAMF1 gene inhibited PRRSV replication significantly and reduced the levels of key signaling pathways, including MyD88, RIG-I, TLR2, TRIF, and inflammatory factors IL-6, IL-1β, IL-8, TNF-β, TNF-α, and IFN-α in vitro. However, the knockdown of the SLAMF1 gene could enhance the replication of the PRRSV and the levels of key signaling pathways and inflammatory factors. Overall, our results identify a new antagonist of the PRRSV, providing a new reference and direction for PRRSV disease resistance breeding.

**Abstract:**

The porcine reproductive and respiratory syndrome virus (PRRSV) causes a highly contagious disease in domestic swine. Signaling lymphocytic activation molecule family member 1 (SLAMF1) is a costimulatory factor that is involved in innate immunity, inflammation, and infection. Here, we demonstrate that overexpression of the SLAMF1 gene inhibited PRRSV replication significantly and reduced the levels of key signaling pathways, including MyD88, RIG-I, TLR2, TRIF, and inflammatory factors IL-6, IL-1β, IL-8, TNF-β, TNF-α, and IFN-α in vitro. However, the knockdown of the SLAMF1 gene could enhance replication of the PRRSV and the levels of key signaling pathways and inflammatory factors. Overall, our results identify a new, to our knowledge, antagonist of the PRRSV, as well as a novel antagonistic mechanism evolved by inhibiting innate immunity and inflammation, providing a new reference and direction for PRRSV disease resistance breeding.

## 1. Introduction

The PRRSV is a highly infectious agent which causes severe reproductive impairment in breeding pigs and respiratory disorders in piglets [1,2,3]. Since the PRRSV was detected in wild boar in Europe and North America in 1987, it has caused huge economic losses to the global pork industry [4]. In vivo, the main cell type of PRRSV infection is monocytes, especially porcine alveolar macrophages (PAMs), which are the main target of the PRRSV infection [5,6]. The PRRSV first binds to specific receptors on the surface of PAMs and then enters eukaryotic cells through endocytosis [7]. The PRRSV releases virions in PAMs and transfers them through the body’s blood and lymphatic circulatory systems, leading to immunosuppression and, eventually, co-infection with viruses or bacteria [8,9], resulting in weakened host adaptation and innate immune response [10,11]. The PRRSV’s invasion causes a series of immune responses, and then virions escape from the host immune system to favor their own replication by interacting with the host. The host immune system takes some measures to suppress the virus replication during the virus invasion. The cellular protein nucleoporin 62 (Nup62) interacts with nsp1β, leading to the inhibition of host antiviral protein expression, revealing a new strategy of immune escape [12]. This is the main cause of PRRSV infection leading to high morbidity and mortality in pigs. Hence, we need to further study the replication mechanism of the PRRSV and analyze the host factors involved in the infection. Despite extensive research, there is currently no effective prevention or treatment to reduce the PRRSV epidemic. Therefore, it is of great significance for animal husbandry to find a standard method to prevent PRRSV transmission effectively.

Signaling lymphocytic activation molecule family member 1 (SLAMF1) is a cell-surface glycoprotein expressed on a variety of immune populations that regulates cell–cell interactions, activation, differentiation, and inflammatory responses [13,14], and has been identified as a potential target for inflammatory diseases. SLAMF1 can activate NF-κB, P38-MAPK, PI3K-Akt, and other signaling pathways through RIG-I and toll-like receptors, thereby promoting the IRF3-mediated secretion of IFNs, TNF- α, and various interleukins [15,16]. In addition, it can also be incorporated into the phagosome with the pathogen, inhibiting the proliferation of the pathogen and accelerating the elimination of the pathogen by promoting the maturation of the phagosome [17]. Therefore, SLAMF1 plays an important role in activating the host antiviral immune response.

Our previous transcriptome study of PRRSV-infected PAMs found that most differentially expressed genes in the phagocytic pathway were down-regulated [18]. In addition, a large number of differentially expressed genes, including TRAM, TLR4, and a variety of cells, were enriched in the immune pathway and autophagosome maturation pathway involved in SLAMF1. At the same time, we detected the absolute viral load in serum and found that the expression of the SLAMF1 gene was inversely proportional to the viral load of the host [18]. These results suggest that PRRSV infection could inhibit the phagocytic maturation pathway of host cells, and the expression level of SLAMF1 was negatively correlated with the viral load of the PRRSV. This study was conducted to investigate porcine SLAMF1 functions in PRRSV proliferation.

The frequent genetic variation of the PRRSV causes the diversity of PRRSV strains, reducing the protection of PRRS commercial vaccines on swine, and posing huge challenges to the prevention and control of PRRS [19,20]. Our study intended to provide a new non-vaccine idea for the prevention and control of the PRRSV. In this study, we overexpressed the exogenous porcine SLAMF1 gene and the down-regulated endogenous SLAMF1 gene in Marc-145 cells to study the effect of the SLAMF1 gene on PRRSV proliferation and its role in inducing an inflammatory response, thus revealing a potential PRRSV resistance gene. These results not only provide an important basis for further research on the interaction between the SLAMF1 gene and the PRRSV but also provide a new reference for PRRSV resistance genetics and breeding.

## 2. Materials and Methods

### 2.1. Cells, Viruses, Vectors, and Antibodies

PRRSV strain HuB1801, eukaryotic expression vector pEGFP-N1, and the Marc-145 cells were preserved in the Institute of Animal Husbandry and Veterinary. The SLAMF1 primary antibody and HRP-labeled goat anti-mouse IgG were purchased from Sanying Biotechnology Co., Ltd. (Wuhan, China).

### 2.2. Construction of SLAMF1 Gene Eukaryotic Expression Plasmid

Primer Premier 6.0 software was used to design specific primers based on the nucleotide sequence of the porcine SLAMF1 gene. The primer sequences are listed in Table 1. The total RNA of Marc-145 cells was extracted by using a Total RNA Kit I (Omega Bio-Tek, Norcross, Georgia) and then reverse transcribed by using a cDNA Synthesis Kit following the manufacturer’s instructions. The cDNA was used as the template for PCR amplification; concisely, 25 μL of 2 × Phanta Mix, 2 μL of cDNA, 2.5 μL of the forward primer, 2.5 μL of reverse primer, and 18 μL of ddH_2_O were added to each well. The procedure for PCR reactions was as follows: PCR activation at 95 °C for 3 min, followed by 35 cycles of 95 °C for 15 s, 55 °C for 15 s, and 72 °C for 40 s. The SLAMF1 gene was digested with *Xma*I and *Xho*I and was then ligated into pEGFP-N1. The recombinant plasmid was named pEGFP-N1-SLAMF1. PCR and Sanger sequencing was utilized for verification.

The endotoxin plasmid can be extracted according to the instructions, and transfection can be carried out when the cell density reaches about 70–80%. Lipofectamine2000 was used to transfect Marc-145 cells and si-RNA according to the instructions.

### 2.3. Real-Time Quantitative PCR (qRT-PCR)

RNA extraction and reverse transcription were performed with TRIzol reagent (Invitrogen, Carlsbad, CA, USA) and a reverse transcription kit (Nanjing Novizan Biotechnology Co., Ltd., Nanjing, China) according to the manufacturer’s instructions. Then, RNA concentration and purity were determined by a Nanodrop 2000 UV spectrophotometer. The real-time quantitative PCR was performed using a 10 μL system and SYBR Green Master Mix (Roche, Shanghai, China) at ABI QuantStudio^TM^ 6. The sequences of qRT-PCR primers used in the experiment are shown in Table 2. Then, qRT-PCR was performed in a 10 μL reaction including 5 μL of SYBR Green Mix, 1 μL of cDNA, 0.2 μL of the forward primer, 0.2 μL of reverse primer, and 3.6 μL of ddH_2_O. For the determination of the relative mRNA expression levels, the following formula was used: ΔΔCT = ΔCT _(treated group)_ − ΔCT _(saline group)_.

### 2.4. Western Blot (WB)

The cells were lysed in the RIPA strong buffer (Beyotime, Shanghai, China) containing a 1% protease inhibitor cocktail and 1% phosphatase inhibitor complex on ice, and were boiled with 10μL SDS-PAGE loading buffer for 10 min and then placed in an ice bath for 20 min. Total protein samples were separated by using SDS-PAGE and transferred electrophoretically onto polyvinylidene fluoride (PVDF) membranes. The membrane was blocked with 5% skim milk overnight. The membrane was probed with SLAMF1 primary antibody (1:5000) and HRP-labeled goat anti-mouse IgG (1:5000) successively. The antibody-specific protein was visualized using an enhanced chemiluminescence (ECL) detection system, and β-actin(1:5000) was used as an internal control.

### 2.5. Statistical Analysis

The results of each group of experiments were independently repeated at least three times. All statistical analyses were performed using GraphPad Prism 8.0.1, and the data are expressed as the mean ± standard error of the mean. The significant differences among groups were determined by one-way or two-way analysis of variance (ANOVA). A multiple *t*-test was used to analyze the significance of the mean difference in each group of data. Differences with *p*-values < 0.05 were considered significant and are designated with * in the figures. Differences with *p*-values < 0.001 were considered significant and are designated with *** in the figures.

## 3. Results

### 3.1. Expression of SLAMF1 in Marc-145 Cells by pEGFP-N1-SLAMF1

The SLAMF1 gene fragment was detected by agarose gel electrophoresis and showed a bright band at 1032 bp (Figure 1a). Double enzyme digestion was used to identify the selected single-colony plasmid, and agarose gel electrophoresis showed two bands consistent with the expected size (Figure 1b), indicating that SLAMF1 had successfully inserted into the PEGFP-N1 vector. The qRT-PCR results indicate that the mRNA expression of SLAMF1 was significantly increased by pEGFP-N1-SLAMF1 (*p* < 0.001) (Figure 1c). The protein expression of the EGFP-labeled SLAM gene was detected by western blot, which was consistent with the predicted protein size (80 kDa), with the band size of the empty vector being about 27 kDa. The WB results also indicate that the protein level of SLAMF1 was increased by pEGFP-N1-SLAMF1 (Figure 1d).

### 3.2. Overexpression of SLAMF1 Inhibits PRRSV Replication

To study the influence of porcine SLAMF1 on the proliferation of the PRRSV, we infected Marc-145 cells that overexpressed SLAMF1 for 24 h with the PRRSV and collected protein samples at multiple time points after PRRSV infection. The results demonstrate that the PRRSV N protein level in cells transfected with pEGFP-N1-SLAMF1 was significantly lower than that in cells not transfected with pEGFP-N1-SLAMF1 at 36 h post-infection (HPI), but not significantly at other time points (Figure 2a). The qRT-PCR results show that SLAMF1 had different inhibitory effects on PRRSV infection at multiple multiplicities of infection (MOI), among which the MOI of 0.1 and 18 hpi show the most significant effect (Figure 2b).

To determine the effect of PRRSV infection on the expression of the SLAMF1 gene in Marc-145 cells, Marc-145 cells were inoculated with the PRRSV at an MOI of 0.1, 0.5, and 1, respectively. The qRT-PCR results show that the expression of the SLAMF1 gene was inhibited by PRRSV infection in cells, and the inhibition possesses an MOI dependence. Marc145 cells were infected by the PRRSV at an MOI of 0.1, and its RNA was collected at 0 h, 12 h, 24 h, 48 h, and 72 h post-infection, respectively (Figure 2c). The results show that the PRRSV inhibits the SLAMF1 mRNA level after 24 hpi and show a time dependence.

### 3.3. Knockdown of SLAMF1 Enhances PRRSV Replication

According to the gene sequence of rhesus monkey SLAMF1, a simian si-SLAMF1 interfering plasmid was designed, among which, si-SLAMF1-197 had the best interference effect (Figure 3(a1)). 24h after transfection into Marc-145 cells, western blot was used to detect the changes in the SLAMF1 gene in cells. The results show that the level of the SLAMF1 in cells was significantly reduced after interfering with SLAMF1 genes (Figure 3(a2)).

To study the effect of interfering with the porcine SLAMF1 gene on pathogen proliferation, western blot was used to detect the expression of the PRRSV N protein. The results show that interfering with the SLAMF1 gene could effectively inhibit the proliferation of the PRRSV for 36h after infection. The expression level of the PRRSV N protein in cells transfected with PEGFP-N1-SLAMF1 was significantly lower than that in cells not transfected with PEGFP-N1-SLAMF1 (Figure 3b).

The qRT-PCR results show that the SLAMF1 gene was significantly down-regulated (*p* < 0.01) after siRNA interference compared with that in the NC control group. Compared with the NC control group, the viral load in the PRRSV-infected group was significantly up-regulated (*p* < 0.01), indicating that interference with the SLAMF1 gene can effectively promote the proliferation of the PRRSV (Figure 3c).

### 3.4. SLAMF1 Gene Overexpression Inhibits PRRSV-Induced Inflammation

qRT-PCR was used to detect the expression levels of key signaling molecules in Marc-145 cells with SLAMF1 expression and PRRSV infection. The results show that overexpression of SLAMF1 inhibited the proliferation of PRRSV, and the levels of key signaling pathways MyD88, RIG-I, TLR2, and TRIF in cells were significantly inhibited (*p* < 0.001) (Figure 4a). The expression levels of cytokines IL-6, IL-1β, IL-8, TFN-β, TNF, and IFN-α were significantly lower (*p* < 0.001) than those in the PRRSV-infected group (Figure 4b).

### 3.5. Knockdown of Porcine SLAMF1 Promotes PRRSV-Induced Inflammation

The qRT-PCR results show that, after si-SLAMF1 promotes the proliferation of the PRRSV, the expression levels of the key signaling pathways TRIF and TLR2 were significantly (*p* < 0.05) higher than those of the PRRSV-infected group, and the expression level of MyD88 was extremely significantly (*p* < 0.01) higher than that in the PRRSV-infected group (Figure 5a). The expression level of IL-6 was significantly (*p* < 0.05) higher than that in the PRRSV-infected group, and the expression level of IFN-α was extremely significantly (*p* < 0.001) higher than that in the PRRSV-infected group (Figure 5b).

## 4. Discussion

The PRRSV has brought huge losses to the pork industry in China and even the world. Although the resistance of pigs can be improved through vaccine research and development, it is difficult to control the PRRSV epidemic simply by using vaccines due to the strong variability of the PRRSV [21]. Therefore, PRRSV resistance genetic breeding has become an important way to control PRRSV infection. We can compare the response differences of PRRSV infection in pigs or cells of various breeds, analyze the pathways of PRRSV infection, screen and verify the antiviral activity of key genes, and breed a variety with disease resistance or tolerance to PRRSV infection, to improve the resistance ability of the pig population to the PRRSV. Interferon lambda receptor 1 (IFNLR1) is a type II cytokine receptor that binds to interleukins IL-28A, IL29B, and IL-29, known as type III IFNs (IFN-λs). IFN-λs play an antiviral role in preventing and curing infection through the JAK-STAT signaling pathway [22,23,24]. IFNLR1 overexpression induces the activation of the JAK/STAT pathway, thus inhibiting the proliferation of PAMs infected by the PRRSV [25]. Other related studies, including the nucleotide-binding oligomer domain-like receptor NLRX1 gene [26], and the interferon-inducible genes OAS1, IFITM1 [27], MX1 [28,29], and IFI30 [30], have also been reported to inhibit PRRSV replication to a certain extent.

SLAMF1 is a co-stimulatory factor expressed on the surface of most immune cells [31,32] and plays an important role in signal transduction between immune cells [33] and the promotion of phagosome maturation [34]. Numerous studies revealed that SLAMF1 plays an important role in natural immunity. In addition, SLAMF1 can activate RIG-I and toll-like receptors and enhance NF-κB, p38MAPK, AKT, PI3K, and other signaling pathways, thus promoting the IRF-3-mediated secretion of IFNs, TNF-α, and various interleukin factors [35,36]. On the other hand, SLAMF1 is related to the migration ability of macrophages, DC cells, and bone marrow cells [17]. Antigen-presenting cells (APCs) such as macrophages enwrap pathogens from phagosomal vesicles by actin rearrangement when pathogens invade the body. Then, they fuse with lysosomes to form phagolysosomes, killing and degrading pathogens in time after the primary phagosomal vesicles mature [37]. During this process, SLAMF1 can enter phagosomes with pathogens simultaneously, mobilizing the ubiquitination enzyme complex and NADPH oxidase NOX2 complex, and accelerating pathogen clearance by promoting phagosomal maturation. However, the role of SLAMF1 in PRRSV infection remains unclear. In this study, we first constructed the porcine SLAMF1 gene into the eukaryotic expression vector and verified the anti-PRRSV function of the porcine SLAMF1 gene by overexpressing the SLAMF1 gene in Marc-145 cells. Our results show that the mRNA expression of the SLAMF1 gene was significantly up-regulated (*p* < 0.001), and the mRNA expression of the virus was significantly inhibited (*p* < 0.01) 36 h post-PRRSV infection.

PRRSV infection induces the expression and release of a large number of inflammatory factors (such as IL-1β, IL-6, IL-8, TNF-α, etc.), leading to acute inflammation in the lungs, ultimately leading to respiratory failure and death [38]. NF-κB is a transcription factor that plays a key role in the regulation of inflammatory and immune responses, as well as cell proliferation and apoptosis. For example, the PRRSV Nsp2 can significantly activate the NF-κB pathway and thus induce the expression of pro-inflammatory factors such as IL-6 and IL-8 [39]. The PRRSV Nsp1α can inhibit LUBAC-induced NF-κB activation and pro-inflammatory cytokine expression in the early stages of PRRSV infection, and PRRSV Nsp1α inhibits NF-κB activation by targeting the LUBAC complex [40]. The innate immune signaling pathway includes PRRs, intermediate junction molecules, downstream signaling molecules of the cascade response, and, ultimately, transcription factors. NLR receptor genes recognize pathogenic microorganisms that invade the cell and regulate the innate immune response of the body against pathogenic microorganisms [41]. SLAMF1 genes act as negative regulators of type I interferon production and NF-κB activation, and SLAMF1 genes knockout mice showed reduced clearance of Gram-negative bacteria [42], indicating that the SLAMF1 gene has a very important role in immune regulation and disease prevention and control. In this study, we examined the expression of PRRs and cytokines in the cellular inflammatory pathway after SLAMF1 inhibited PRRSV replication and revealed that SLAMF1 could exert antiviral effects and suppress the level of virus-induced inflammation in Marc-145 cells.

In this study, we demonstrated that the expression levels of inflammatory factors (IL-6, IL-1B, IL-8, TFN-β, TNF, and IFN-α) were significantly inhibited by SLAMF1. To further reveal the role of SLAMF1 in inhibiting PRRSV-induced inflammation, we knocked down the level of the SLAMF1 gene by siRNA. The results show that the interference of endogenous SLAMF1 in Marc-145 cells promoted the proliferation of the PRRSV, and the key signal molecules such as MyD88, TRIF, RIG-I, and TLR2. The expression levels of inflammatory factors such as IL-1β, IL-6, IL-8, IFN-β, and TNF-α also increased correspondingly, suggesting that si-RNA interference with the SLAMF1 gene promotes cellular inflammation caused by PRRSV infection. Overall, our study provides a theoretical basis for the research and application of SLAMF1 and provides a new reference and direction for PRRSV disease resistance breeding.

## 5. Conclusions

We conclude that the eukaryotic expression vector of the porcine SLAMF1 gene was successfully constructed and it could be stably expressed in Marc-145 cells. Overexpression of the porcine SLAMF1 gene in Marc-145 cells could effectively inhibit PRRSV proliferation and down-regulate PRRSV-induced cellular inflammatory response, while the si-SLAMF1 gene had the opposite effect. Moreover, PRRSV infection of Marc-145 cells could inhibit the expression of the SLAMF1 gene in host cells.

## Figures and Tables

**Figure 1 animals-12-03542-f001:**
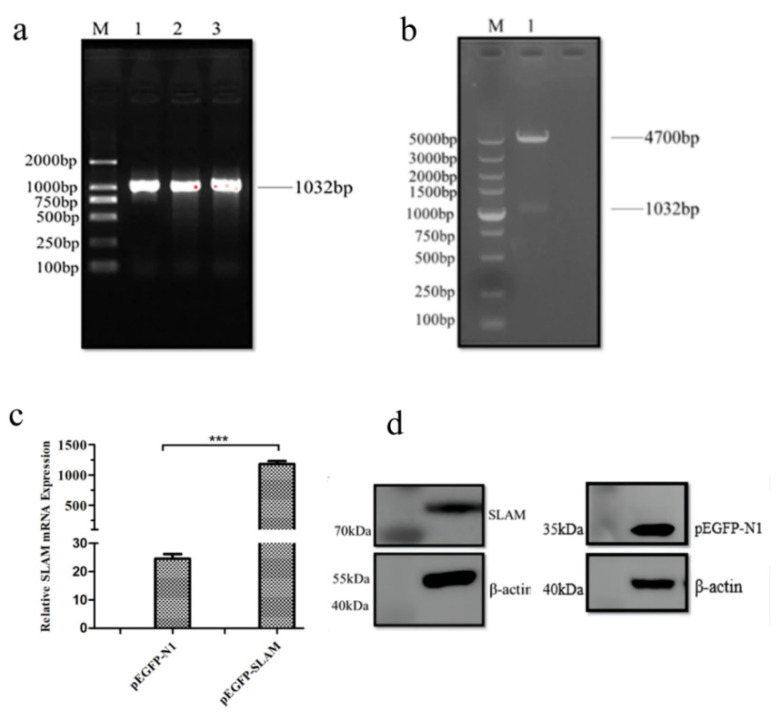
pEGFP-N1-SLAMF increases SLAM level. (**a**) The PCR product of the SLAM gene. M: DL2000 Marker; Line1~3: gene fragment of SLAM. (**b**) Double enzyme digestion product of pEGFP-N1-SLAM. M: DL5000 Marker; 1: The result of pEGFP-N1-SLAM digested with a restriction enzyme. (**c**) Relative SLAM mRNA level was determined by qRT-PCR; Note: the pEGFP-SLAM group compared with the pEGFP-SLAMF1 group: *** means extremely significant difference (*p* < 0.001). (**d**) SLAM protein was determined by western blot.

**Figure 2 animals-12-03542-f002:**
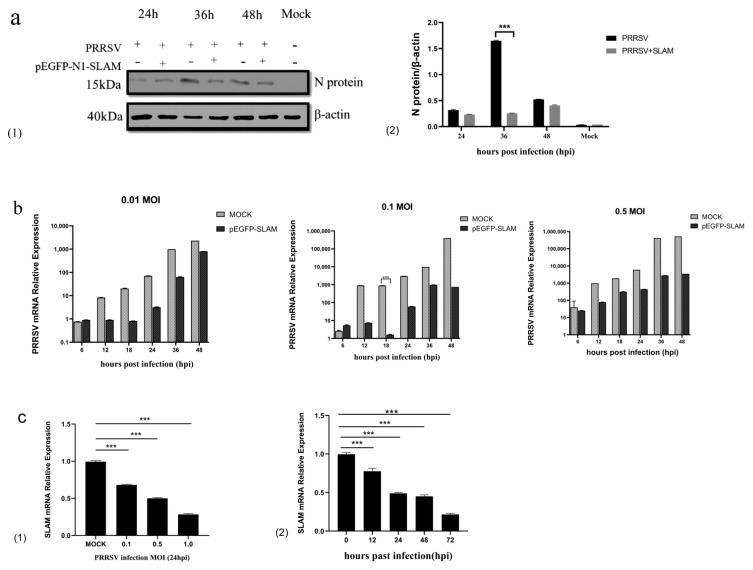
Overexpression of SLAM inhibits PRRSV replication. (**a1**) Expression of PRRSV−N protein at different time points after PRRSV infection. (**a2**) The protein expression of SLAM in infected Marc-145 cells at different time points; (**b**) Effect of overexpression of SLAM and infection with different MOI of PRRSV on PRRSV proliferation at different time points. (**c**) qRT-PCR for detecting the differential expression multiple of SLAM gene at different time points and different titers after PRRSV infection at 24 h. (**c1**) Differential expression multiple of PRRSV-infected 24 h SLAM gene with 0.1–1 MOI. (**c2**) 0.1 Differential expression multiple of SLAM gene 0–72 h after MOI PRRSV infection. Note: *** means extremely significant difference (*p* < 0.001).

**Figure 3 animals-12-03542-f003:**
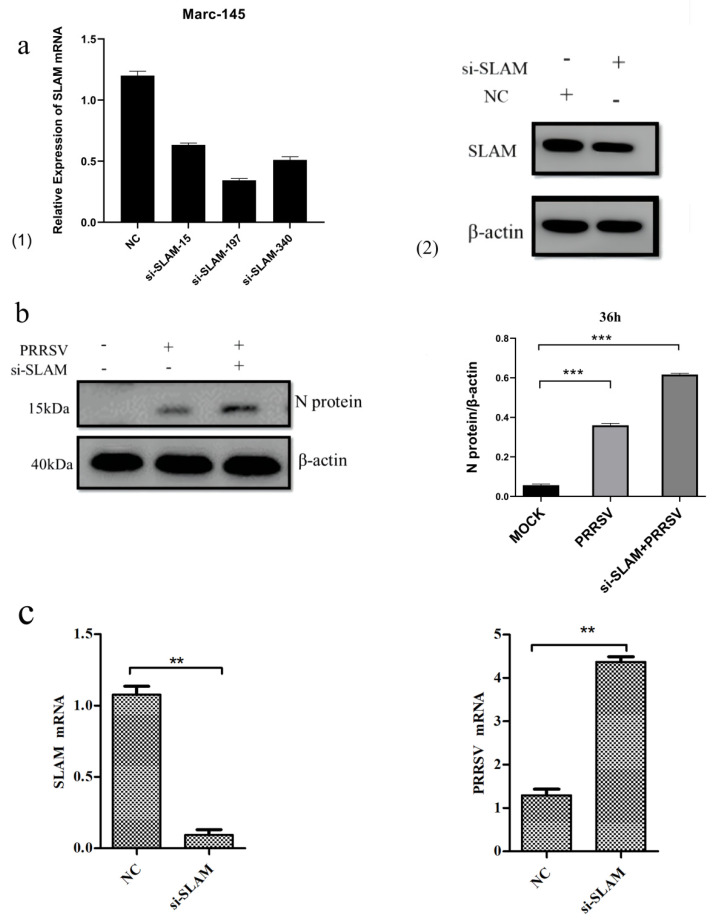
Knockdown of SLAM enhances PRRSV replication. (**a1**) Different synthetic si-SLAM interfering plasmids were transfected into Marc-145 cells, and the mRNA level of the SLAM gene was detected by qRT-PCR 24 h later. (**a2**) Si-SLAM-197 was transfected into Marc-145 cells, and the changes in the SLAM gene in the cells were detected by western blot 24 h later. (**b**) Effect of SLAM gene on PRRSV proliferation was detected by western blot. (**c**) Effect of siRNA interference SLAM gene on PRRSV proliferation was detected by qRT-PCR. Note: ** means significant difference (*p* < 0.01), *** means extremely significant difference (*p* < 0.001).

**Figure 4 animals-12-03542-f004:**
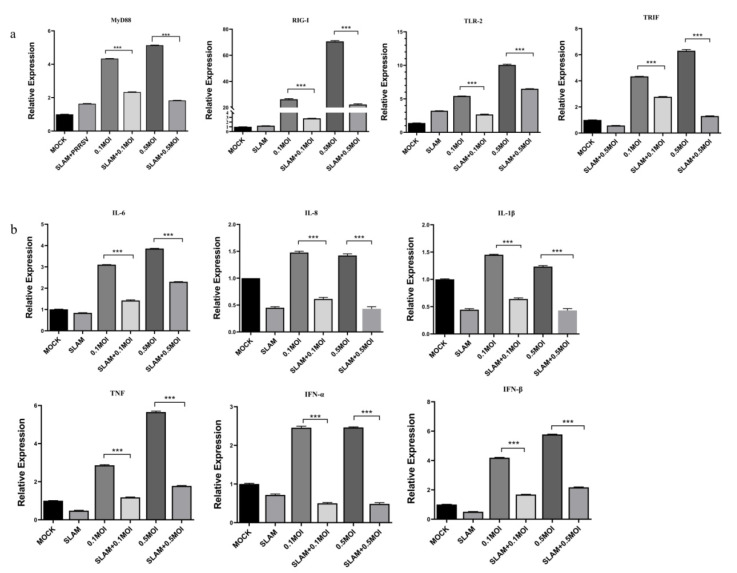
SLAM gene overexpression inhibits PRRSV-induced inflammation. (**a**) Influence of SLAM overexpression on key signaling molecules after PRRSV infection. (**b**) Effects of SLAM overexpression on inflammatory cytokines after PRRSV infection. Note: *** means an extremely significant difference (*p* < 0.001).

**Figure 5 animals-12-03542-f005:**
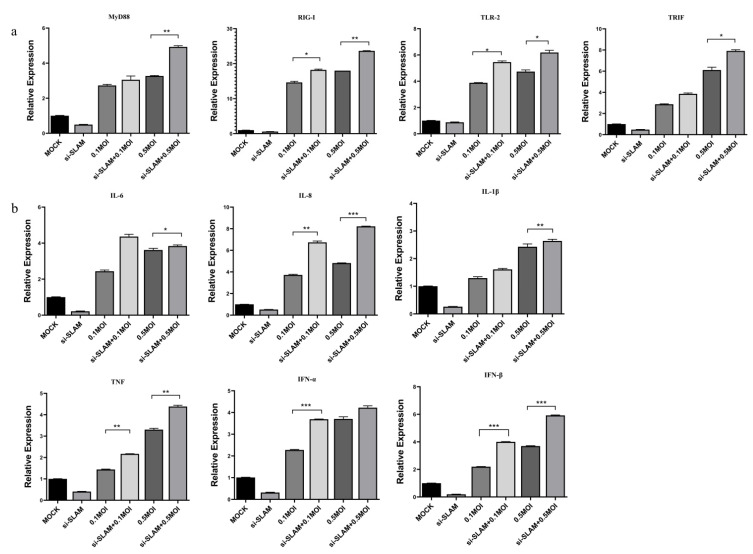
Knockdown of porcine SLAM promotes PRRSV-induced inflammation. (**a**) The effect of si-SLAM on key signaling molecules after PRRSV infection. (**b**) The effect of the si-SLAM gene on inflammatory factors after PRRSV infection. Note: * means significant difference (*p* < 0.05); ** means significant difference (*p* < 0.01); *** means an extremely significant difference (*p* < 0.001).

**Table 1 animals-12-03542-t001:** Primers for SLAMF1 gene expression.

Primer Name	The Necleotide Sequence (5′-3′)	Product Length
SLAMF1-F	FCCGCTCGAGATGCATAAACTAGACAGTAGAGGCA	1032 bp
SLAMF1-R	CCAAGCTTGCTCTCCGGAAGAGTCACG	

**Table 2 animals-12-03542-t002:** Primer for mRNA quantitation used in qRT-PCR.

Primer	Sequences (5′-3′)
mTNF-F	CACCACGCTCTTCTGTCTGCT
mTNF-R	CAGGCTTGTCACTTGGGGTT
mIL-6-F	ACTGGTCTTTTGGAGTTTGAGG
mIL-6-R	GCTGGCATTTGTGGTTGGTT
mRIG-I-F	TGATTGCCACCTCAGTTG
mRIG-I-R	TTCCTCTGCCTCTGGTTTG
mIFN-α-F	CATTGCCCTTTGCTTTACTGAT
mIFN-α-R	CTGGAGCCTTCTGGAACTGGT
mIL-1B-F	TCCCACGAGCACTACAACGA
mIL-1B-R	CTTAGCTTCTCCATGGCTACAACA
mIL-8-F	CTGGCGGTGGCTCTCTTG
mIL-8-R	CCTTGGCAAAACTGCACCTT
mTLR2-F	CTGCAAGCTGCGGAAGATAAT
mTLR2-R	TTCCTGCCGAGCCTCATC
mTRIF-F	ACTCGGCCTTCACCATCCT
mTRIF-R	GGCTGCTCATCAGAGACTGGTT
mMyD88-F	GGCAGCTGGAACAGACCAA
mMyD88-R	GGTGCCAGGCAGGACATC
mIKBa-F	TCCACTTGGCGGTGATCA
mIKBa-R	ATCACAGCCAGCTTCCAGAAG
mGAPDH-F	TGGGGAAGGTGAAGGTCGG
mGAPDH-R	TCCTGGAAGATGGTGATGGG
qORF7-F	TCAGCTGTGCCAAATGCTGG
qORF7-R	AAATGGGGCTTCTCCGGGTTTT

## Data Availability

The datasets used and/or analyzed during the current study are available from the corresponding author on reasonable request.

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
