# Peer review of "Signaling Lymphocytic Activation Molecule Family Member 1 Inhibits Porcine Reproductive and Respiratory Syndrome Virus Replication"

_animals, 2022, doi:10.3390/ani12243542_

Round 1

Reviewer 1 Report

High genetic variability in the PRRSV genome is a major constraint in the design and development of a broadly protective vaccine.  PRRSV remains a significant threat to the pig industries worldwide. In this manuscript by Song et al, the authors have investigated the role of SLAMF1 in PRRSV propagation and pathogenesis. Below are my concerns and comments on this manuscript.

Concern #1.

SLAMF1 has been identified as one of the inducers of innate immune responses via NF-kB, P8-MAPK, and PI3K-AKT, and authors in this manuscript have shown that PRRSV suppresses the level of SLAMF1. So, my question here is how suppression of this SLAMF1 gene expression during PRRSV infection enhances these responses instead of suppressing them?. The author should comment on this observation.

Concern #2.

Introduction: Brief information about the known signaling pathways targeted by PRRSV to escape host innate immunity will help readers to appreciate the importance of this work.

Discussion: Authors should have compared their current research results with the published research on SLAMF1.

References: Check for proper references, some are not appropriate as they are cited.

Minor concern:

Line 51: Please provide a reference for this sentence.

Line 57: This sentence has been repeated twice, please correct this.

Line 93: You mean endotoxin free?

Line 115: SLAMF1 and b-actin antibody dilutions are missing.

Line 191-193: This interpretation of Figure 3c does not match with results. Please correct this.

Figure 3b: Please correct the figure, I believe you are missing si-SLAMF1 at the 36hr plot.

Figure 5: Please correct the figure, I believe you are missing si-SLAMF1 in the TNF plot. Also, check figure legend

Reviewer 2 Report

The manuscript entitled Signal lymphocyte activation molecule family member 1 inhibit Porcine reproductive and respiratory syndrome virus Replicationstudied the roles of SLAMF1 in PRRSV infection and replication. This work provided some useful information on the development of antagonists of PRRSV. There are still some improvements that must be made in the manuscript.

1. The manuscript must be carefully examined and corrected for spelling errors and formatting of paragraphs. It is best to have the manuscript English polished by a professional or native English speaker.

2. Please carefully check the abbreviations of proper nouns in the text, provide the full names when they first appear in the manuscript, and unify them. For example: SLAMF1 and SLAM, TFN and TNF, IL and Interleukin, siRNA and si-RNA.

3. In Fig2b, this study explored the effect of SLAMF1 expression on PRRSV infection. Please supplement the significance analysis of the relevant results and label them in the bar chart.

4. The authors studied the replication and inflammation level of PRRSV under the circumstance of SLAMF1 overexpression and SLAMF1 knockdown and found that SLAMF1 overexpression inhibited PRRSV replication and PRRSV-induced inflammation, while SLAMF1 knockdown promoted PRRSV replication and PRRSV-induced inflammation. What makes me wonder is whether the change in PRRSV replication level is related to the level of PRRSV-induced inflammation.

Reviewer 3 Report

1. “These results suggested that PRRSV infection could inhibit the phagocytic maturation pathway of host cells, and the expression level of SLAMF1 was negatively correlated with the viral load of PRRSV.” There is duplication in lines 57-61. to our knowledgein lines 20-21? and too many mistakes in the manuscript.

 2. The MARKER size in Figure 1 is incorrectly marked

 3. The sequence of pictures in lines 189, 194 and 199 is wrong

 4. Please confirm whether the abscissa of group MyD88 and group TRIF in Figure 4 is wrong.

 5. FIG. 5 Group IL-6 is incorrectly marked

 6. The first paragraph in the discussion is too long. I suggest the author reduce the length and put it in the introduction. There is too little content related to this study in the discussion. Please make a comprehensive analysis based on earlier research results.

 7. The very complex typing of PRRSV is an important feature of the virus. Which branch does the HuB1801 strain used in this paper belong to?

 8. Where do the N protein antibodies come from?

 9. The diagram notes of Fig2a are not clearly described. In the PRRSV-infected group shown in Figure 2b, the amount of N protein at 48hpi is significantly lower than that at 36hpi, which seems abnormal.

 10. Figure 2: The result of Figure 2b is how long after infection with PRRSV is measured? Because you mentioned earlier that the timing of the infection is important. Line 183, “The results showed that PRRSV inhibits SLAMF1 mRNA level after 24hpi, and showed a time-dependence”, the previous results show that SLAMF1 has the most obvious effect on PRRSV at 18 hours, and you did not measure the effect of PRRSV on SLAMF1 at 18 hours, but directly said that the impact is the highest at 24 hours, which is not scientific. Since it could have occurred at any point between 12 and 24 hours, especially at 18 hours, your results also show that there is no significant difference between 24 hours and 48 hours.

11. Both Marc-145 and PAM cells are completely different cells. Many studies showed that the results in Marc-145 could not repeat in PAM, and the difference was even greater in vivo. So, in this study, how is the relationship of SLAMF1 inhibiting PRRSV Replication in Marc-145, PAM and in vivo.
